# Investigation of Compression Strength and Heat Absorption of Native Rice Straw Bricks for Environmentally Friendly Construction

**Suchat Tachaudomdach \* and Sakda Hempao**

Department of Industrial Technology, Faculty of Science and Technology, Chiang Mai Rajabhat University, Chiang Mai 50300, Thailand
\* Correspondence: suchat_tac@cmru.ac.th

**Abstract:** The purpose of this study was to compare the efficiency of native rice straw mixed bricks, prototype bricks and brick blocks that are sold in the market. The comparison looked at four aspects, as follows: (1) compression strength, (2) heat absorption, (3) weight of the brick and (4) price. In this study, the native rice straw species from the Pa Pae sub-district, Mae Sariang District, Mae Hong Son Province, Thailand, were used to replace cement, sand and stone in 200, 300, 400 and 500 g amounts. The ingredients were then mixed and put into a hand-pounded mold. When the extrusion was finished, it was removed from the mold and cured for 7, 14 or 28 days. Brick block performance was then tested. The results showed that 200 g of native rice straw mixed with cement brick had the best performance in all four aspects. The 200 g native rice straw mixed with cement brick was able to bear the weight of 1.26 kg/cm$^2$. It had the best heat transfer and was able to reduce the temperature inside the brick-block construction by 10 degrees Celsius. Its weight per brick was 7.59 kg and the price was only 3.17 baht ($0.09 USD) per brick. In conclusion, the native rice straw mixed bricks had low thermal conductivity, are attractive for energy saving when used as wall insulation and are suitable for environmentally friendly construction.

**Keywords:** compression strength; heat absorption; brick; native rice straw brick; environmentally friendly construction

## 1. Introduction

Nowadays, construction often uses recycled materials from agriculture as they are the most cost-effective and efficient. This leads to a reduction in waste and a reduction in environmental impact [1]. The construction consists of two main parts: (1) building materials themselves and (2) parts relating to the construction of buildings or structures from the production of materials for use in construction, such as materials that are structural components of buildings, including cement, steel or wood, and architectural components such as roofs, ceilings, tiles, bricks, etc. [2].

The objective of this study was to develop a brick mixed with native rice straw. This study will focus on research studies to obtain bricks of the same quality as the standard, with ingredients derived from local natural materials coming from agricultural waste. Using these materials reduces burning, which in turn reduces air pollution problems and allows builders to use materials from within their own community or locality. The use of these local materials can be further developed into products for sale, generating income for local communities as well.

The objectives of this study were (1) to compare the compression strength of commercially produced bricks and bricks mixed with native rice straw; (2) to compare the heat transfer coefficient between commercially produced bricks and bricks mixed with native rice straw; (3) to compare the weight between commercially produced bricks and bricks

mixed with native rice straw; and (4) to compare the price between commercially produced bricks and bricks mixed with native rice straw.

## 2. Literature Review

### 2.1. Brick Blocks

Brick blocks or concrete blocks are gray in color. The distinguishing feature of brick blocks is a vertical hollow or hole in the middle, which is considered an advantage. This allows for good ventilation and heat transfer. Concrete blocks are popular because they are cheap and convenient for construction, and construction can be done quickly due to the large size of the bricks. However, if the plaster is not standardized, there may be leakage. Further, a lack of standardization in materials means these bricks may be able to bear less weight than other types of bricks, and they are unsuitable for drilling for mounting or hanging devices [2].

Concrete blocks can be classified into two types, as follows:

1.  Load-bearing concrete blocks, as shown in Figure 1. These blocks have the appearance of a smooth bar with a vertical hole in the middle. They are commonly used to make walls.

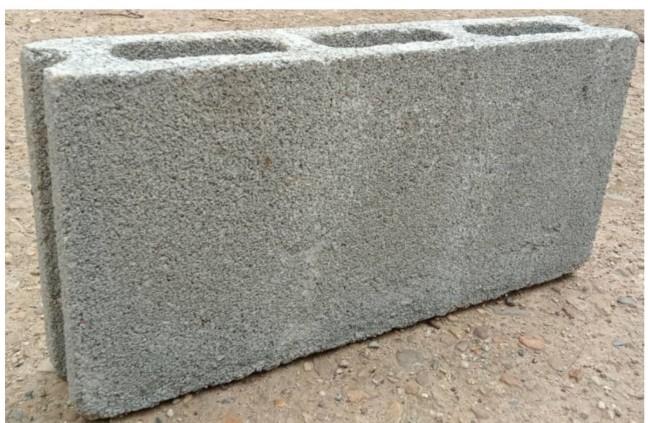

**Figure 1.** Load-bearing concrete block.

2.  Non-load-bearing concrete block, as shown in Figure 2. These are patterned blocks. Once formed, a pattern is formed that can be seen, and sunlight and wind move through the block. Villagers in Mae Hong Son Province often refer to these as "brick block vents".

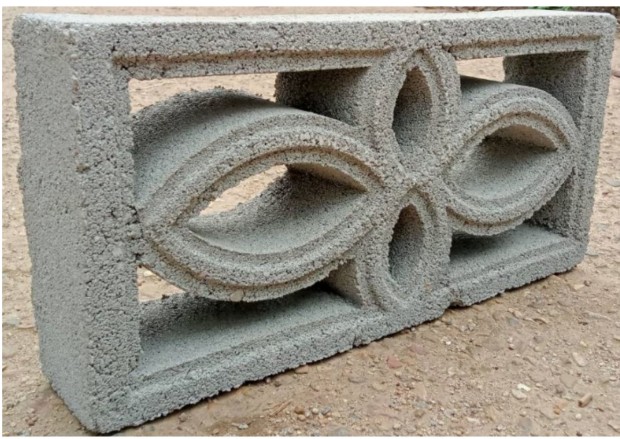

**Figure 2.** Non-load-bearing concrete block.

### 2.2. Standards of Brick Block Production

Nowadays, the production of brick blocks in Thailand conforms to industrial product standard 57-1987, which is the industrial product standard on concrete blocks, and to industrial product standard 58-1987, which is the industrial product standard on non-loaded concrete blocks. Both standards designate that brick blocks be rectangular and approximately $7 \times 19 \times 39$ cm [3]. Raw materials used in the production of brick blocks include Portland cement type 1, dust stones 0.01–0.3 mm in size, sand and water [4].

### 2.3. Building Walls

House walls can be both external and internal, and the exterior walls protect the house from various weather conditions, including sunlight, heat and/or wind. The interior walls act to divide the living space in the house into unique areas, such as the bedroom, living room, kitchen, etc. In addition to dividing the space into external and internal areas, the walls are also divided into external and interior walls. Walls may also be structural or load-bearing walls, which are broken down into reinforced concrete load-bearing walls and load-bearing walls that use full-panel masonry. There are two characteristics of masonry walls: show-themed masonry and plastered masonry walls. Construction of load-bearing walls is more expensive than other types of walls [3].

### 2.4. Characteristics of Wall Building Materials

Compression strength is defined as the tested resistance of a material for calculating compression, measured in kilograms per square centimeter ($kg/cm^2$) [3].

Weight is the amount of all substances that make up the object, which is constant regardless of the position of the Earth, measured in grams (g) or kilograms (kg) [5].

Heat transfer refers to the thermal conductivity of an object. Heat is transferred from areas of higher temperature to areas of lower temperature and measured in watts per meter-Kelvin [3].

The compression strength of cement bricks is based on cement, sand, and aggregate in the content of grit with various ratios. The selection of the best mixture of brick criteria depends on achieving minimum compression strength according to requirements and then maximizing the proportion of rice straw added. Previous studies have indicated that the quantity of cement in the mixture of the cement bricks affects the compression strength of the bricks, since cement may be responsible for efficiency in the brick mixture. The quantity of sand in the brick enhances the compression declines due to reducing the concentration of cement, which is the binder in cement–sand bricks. Therefore, the quantity of cement added in the mixture of cement bricks is directly proportional to the compression strength [6,7]. Moreover, previous studies found that native rice straws are ecologically friendly and decrease the heat absorption, as well as providing economic advantages [8].

### 2.5. Straw

Straw is the dry stem of rice, which is a byproduct of harvesting rice. When farmers finish harvesting rice, they leave the remnants of the rice plant, also known as straw, in the fields. Some farmers harvest the remaining straw because leaving it may lead to fires. Fires can lead to adverse growing conditions, and the soil in the burned area will degrade. Fires cause changes in soil structure; it becomes coagulated, hardens and loses organic matter, including microorganisms, earthworms or insects. Rice straw can be very useful in handicrafts or can be reintroduced as mulch to provide nutrients for agricultural use. Rice stems and detritus can also be plowed into the soil to increase minerals in the soil. However, there is currently much research on the use of straw as an architectural component to be used in building materials to reduce air pollution caused by the burning of rice paddies [9].

In Thailand, there is an estimated 65 million acres of rice grown each year, producing 24 million tons of rice and straw a year. This straw is used for many purposes, such as fertilizer, mulch, paper, straw mushroom cultivation or even fodder in times of famine [9]. The remaining straw is burnt as an easy and low-cost method of disposal, but it has a negative

impact on the environment by causing smoke containing carbon monoxide (CO), carbon dioxide ($CO_2$), nitrous oxide ($N_2O$) and sulphur oxides ($SO_2$). Consequently, rice straw is an available cheap material that used in construction as mud houses in rural areas [7]. In addition, the seasonal rice straw burning that takes place in Egyptian governorates is an environmental disaster known as the "Black Cloud" [10]. In China, rice straw is used to make bricks to reduce energy consumption and waste from rice harvesting [8]. Furthermore, producing sustainable cement bricks based on heat absorption with minimum compression strength for non-load bearing walls will lead to conserving energy.

*2.6. Related Research*

Previous research has looked at the development and study of the load-bearing properties of concrete blocks made with straw. Researchers have developed concrete blocks with rice straw as an ingredient. They have used cement, stone dust, straw and water in different ratios to build the bricks and then test the strength properties with a compression strength tester to determine the ideal ratio of ingredients of the sample material cubes. The results showed that the compressive strength of a standard industrial product 58-1990 was related to the ratio of cement content (600 g). Stone dust (2400 and 3000 g), straw (500 g) and water accounted for 40% of the cement weight [4].

Previous research also looked at the development of concrete blocks that do not bear weight, according to industry product standards 58-1987, but still increased efficiency in reducing heat conduction into the building by using hyacinth in the concrete block mixture and using a cube shape of $10 \times 10 \times 10$ cm. Researchers compared hyacinth mixtures with compressive resistance meeting industrial product standard 58-1987 to produce test bricks. These were then tested for comparative properties, and researchers found a mixture of ingredients with the lowest thermal conductive coefficient properties. These bricks had compression resistance and moisture content in line with industrial product standard 58-1987. They found concrete blocks with mortar:sand:stone:hyacinth mix rates of 1:3:4.925:0.075 gave a minimum thermal conductivity coefficient of 0.111 watts per meter-Kelvin, lower than the typical market-sold concrete block, which has a thermal conductivity coefficient of 0.189 watts per meter-Kelvin [11].

In addition, previous study also indicated that when the percentage of straw increases in straw cement-based building materials, the compressive strength of these materials decreases; therefore, composite-type building materials with a straw content of no greater than 5% can be used for non-load-bearing structures. During preparation, straw cement-based building materials have problems such as difficulty in stirring and slow hardening. A previously study found that modification of straw cement-based building materials can improve the compressive strength. The results of previous studies were similar to this study, finding that increasing the amount of rice straw reduced the compressive strength [12].

Finally, previous research looked at the development of low thermal conductivity cement blocks from rice straw by considering different mix ratios. Seven sample cubes were produced based on local manual manufacturing processes. The study consisted of measuring thermal conductivity, compressive strength and water absorption, and the results showed that passing rice straw through a 4.75 mm sieve into the mixture could reduce the thermal conductivity and weight of the block. The optimum ratio of soil:cement:sand was 3:1:1 (by volume) at a rice straw ratio of 20%. The properties of the cement block are as follows: thermal conductivity: 0.489 watts per meter-Kelvin; compressive strength: 56.324 $kg/cm^2$; density value: 1706 $kg/m^3$; and water absorption: 17.650%. The rice straw cement block had low thermal conductivity, which is attractive for energy saving when used as wall insulation [13].

A previous study compared the weight of bricks and price of the bricks containing a mixture of native rice straw and the prototype brick. This study found that native rice straws are environmentally friendly and reduce the heat absorption, as well as having social and economic benefits.

### 3. Methodology

Researchers prepared the material by collecting rice straw from Tambon Pa Pae, Mae Sariang District, Mae Hong Son Province, Thailand. This is a native rice variety that is dried in the sun to prevent mold when used in the preparation of brick blocks. Brick blocks consisted of cement, sand and stone dust. After mixing and allowing blocks to form, strength, weight, heat transfer and price were compared. Native rice straw bricks are shown in Figure 3. Finally, the required values were tested and compared with the non-straw-containing bricks (Figure 4).

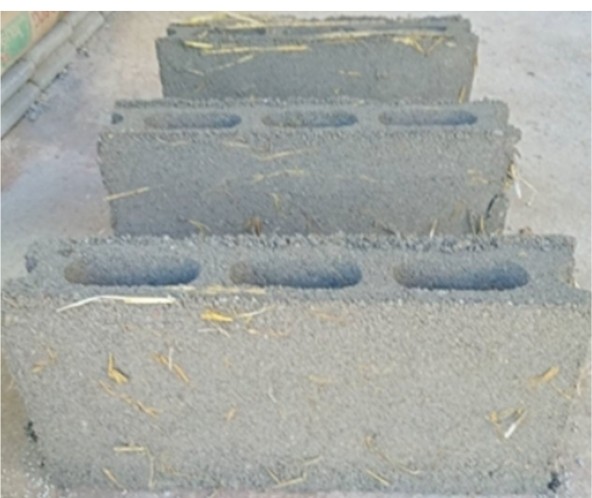

**Figure 3.** Native rice straw bricks.

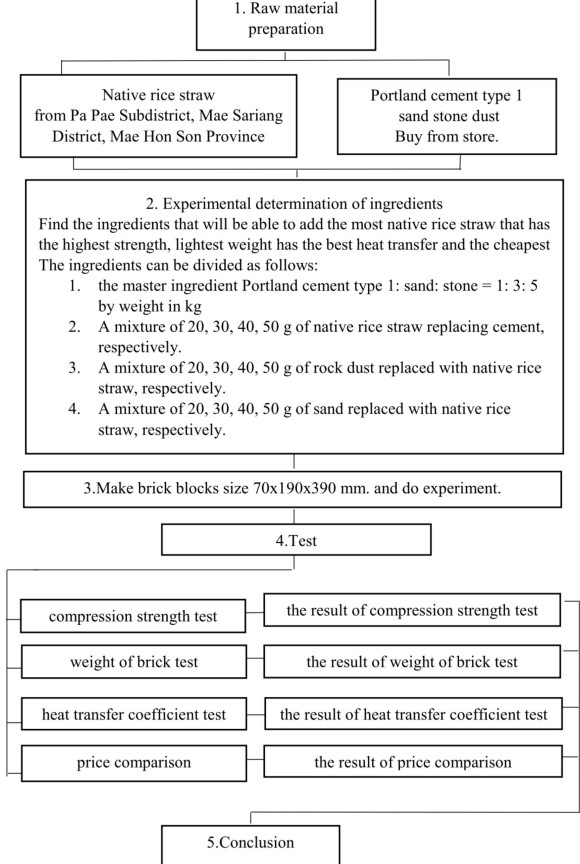

**Figure 4.** Research process.

#### 4. Result

*4.1. Compression Strength*

Prototype brick blocks and brick blocks containing 200, 300, 400 or 500 g of native rice straw in place of cement were made. Each sample was incubated for a period of 7, 14 or 28 days, and then the compression strength of each brick was tested using a compressor. Results of the compression test are shown in Tables 1–3.

**Table 1.** The results of the compression strength test on brick blocks containing 200, 300, 400 or 500 g of native rice straw in place of cement compared to prototype brick blocks.

| Curing Time | Block Type | Compression Strength of Brick (kg/cm$^2$) | | | | | Average |
|---|---|---|---|---|---|---|---|
| | | 1 | 2 | 3 | 4 | 5 | |
| - | Market-sold concrete block | 2.10 | 2.30 | 2.70 | 3.00 | 3.50 | 2.72 |
| | Prototype brick blocks | 1.40 | 1.50 | 1.70 | 1.70 | 1.60 | 1.58 |
| | containing 200 g native straw | 1.30 | 1.20 | 1.10 | 1.30 | 1.00 | 1.18 |
| 7 days | containing 300 g native straw | 1.30 | 1.20 | 1.20 | 1.00 | 1.00 | 1.14 |
| | containing 400 g native straw | 1.00 | 0.90 | 1.00 | 1.10 | 0.80 | 0.96 |
| | containing 500 g native straw | 0.70 | 0.50 | 0.50 | 0.70 | 0.60 | 0.60 |
| | Prototype brick blocks | 1.50 | 1.50 | 1.10 | 1.20 | 1.20 | 1.30 |
| | containing 200 g native straw | 1.30 | 1.30 | 1.00 | 1.00 | 1.00 | 1.12 |
| 14 days | containing 300 g native straw | 0.90 | 1.00 | 1.30 | 1.00 | 1.10 | 1.06 |
| | containing 400 g native straw | 0.80 | 0.90 | 1.00 | 1.10 | 0.90 | 0.94 |
| | containing 500 g native straw | 0.60 | 0.70 | 0.60 | 0.50 | 0.60 | 0.60 |
| | Prototype brick blocks | 1.40 | 1.30 | 1.30 | 1.20 | 1.30 | 1.30 |
| | containing 200 g native straw | 1.00 | 1.50 | 1.00 | 1.50 | 1.30 | 1.26 |
| 28 days | containing 300 g native straw | 1.10 | 1.30 | 1.10 | 1.30 | 1.30 | 1.22 |
| | containing 400 g native straw | 1.30 | 1.20 | 1.30 | 1.20 | 1.00 | 1.20 |
| | containing 500 g native straw | 0.80 | 0.90 | 0.60 | 0.80 | 0.80 | 0.78 |

**Table 2.** The results of the compression strength test on brick blocks containing 200, 300, 400 or 500 g of native rice straw in place of sand compared to prototype brick blocks.

| Curing Time | Block Type | Compression Strength of Brick (kg/cm$^2$) | | | | | Average |
|---|---|---|---|---|---|---|---|
| | | 1 | 2 | 3 | 4 | 5 | |
| - | Market-sold concrete block | 2.10 | 2.30 | 2.70 | 3.00 | 3.50 | 2.72 |
| | Prototype brick blocks | 1.40 | 1.50 | 1.70 | 1.70 | 1.60 | 1.58 |
| | containing 200 g native straw | * | * | 1.50 | 1.00 | 1.50 | * |
| 7 days | containing 300 g native straw | * | * | 0.60 | 0.80 | 1.00 | * |
| | containing 400 g native straw | * | * | * | * | * | * |
| | containing 500 g native straw | * | * | * | * | * | * |
| | Prototype brick blocks | 1.50 | 1.50 | 1.10 | 1.20 | 1.20 | 1.30 |
| | containing 200 g native straw | * | * | * | * | * | * |
| 14 days | containing 300 g native straw | * | 0.70 | 0.80 | 0.80 | 0.40 | * |
| | containing 400 g native straw | * | * | * | * | * | * |
| | containing 500 g native straw | * | * | * | * | * | * |
| | Prototype brick blocks | 1.40 | 1.30 | 1.30 | 1.20 | 1.30 | 1.30 |
| | containing 200 g native straw | * | * | * | * | * | * |
| 28 days | containing 300 g native straw | * | * | * | * | * | * |
| | containing 400 g native straw | * | * | * | * | * | * |
| | containing 500 g native straw | * | * | * | * | * | * |

Note * means the sample brick was in a condition that could not be tested.

**Table 3.** The results of the compression strength test on brick blocks containing 200, 300, 400 or 500 g of native rice straw in place of stone compared to prototype brick blocks.

| Curing Time | Block Type | Compression Strength of Brick (kg/cm²) | | | | | Average |
|---|---|---|---|---|---|---|---|
| | | 1 | 2 | 3 | 4 | 5 | |
| - | Market-sold concrete block | 2.10 | 2.30 | 2.70 | 3.00 | 3.50 | 2.72 |
| | Prototype brick blocks | 1.40 | 1.50 | 1.70 | 1.70 | 1.60 | 1.58 |
| | containing 200 g native straw | * | * | * | * | * | * |
| 7 days | containing 300 g native straw | * | * | * | * | * | * |
| | containing 400 g native straw | * | * | * | * | * | * |
| | containing 500 g native straw | * | * | * | * | * | * |
| | Prototype brick blocks | 1.50 | 1.50 | 1.10 | 1.20 | 1.20 | 1.30 |
| | containing 200 g native straw | * | * | * | * | * | * |
| 14 days | containing 300 g native straw | * | * | * | * | * | * |
| | containing 400 g native straw | * | * | * | * | * | * |
| | containing 500 g native straw | * | * | * | * | * | * |
| | Prototype brick blocks | 1.40 | 1.30 | 1.30 | 1.20 | 1.30 | 1.30 |
| | containing 200 g native straw | * | * | * | * | * | * |
| 28 days | containing 300 g native straw | * | * | * | * | * | * |
| | containing 400 g native straw | * | * | * | * | * | * |
| | containing 500 g native straw | * | * | * | * | * | * |

Note * means the sample brick was in a condition that could be tested.

Based on the results of the compression strength testing, brick blocks containing 200, 300, 400 or 500 g of native straw in place of cement were likely to decrease in compression strength. In ingredient mixtures with a higher proportion of rice straw, bricks had lower compression strength than in ingredient mixtures with a lower proportion of native straw. The compression strength compared to the prototype brick block was reduced. It is evident that the brick blocks that are sold at the market have a greater compression strength than any others tested.

Researchers completed the compression strength testing on 5 samples of brick blocks containing native straw in the amount of 200, 300, 400 or 500 g in place of sand. Weight could not be loaded due to the cement:sand:stone ratio being set at 1:3:5. Therefore, the sand amount was reduced to three times the amount of cement and had to be replaced with native straw in the same amount as the reduced sand. These bricks were not strong, and compared to the brick blocks that are sold on the market, it was evident that commercial brick blocks were able to carry more weight.

Researchers completed the compression strength testing on 5 samples of brick blocks containing native straw in the amount of 200, 300, 400 or 500 g in place of stone. Weight could not be loaded due to the cement:sand:stone ratio being set at 1:3:5. Therefore, the stone amount was reduced to three times the amount of cement and had to be replaced with native straw in the same amount as the reduced stone. These bricks were not strong, and compared to the brick blocks that are sold on the market, it is evident that commercial brick blocks were able to carry more weight.

*4.2. Results of Brick Block Weight Testing*

Prototype brick blocks and brick blocks containing 200, 300, 400 or 500 g of native rice straw in ratio to cement, stone, and sand were incubated for a period of 7, 14 or 28 days. After the incubation period, the brick blocks were weighed, as shown in Tables 4–6.

**Table 4.** The results of the weight of brick blocks containing 200, 300, 400 or 500 g of native rice straw in place of cement compared to prototype brick blocks.

| Curing Time | Brick Type | Weight of Brick (kg) | | | | | Average |
|---|---|---|---|---|---|---|---|
| | | 1 | 2 | 3 | 4 | 5 | |
| - | Market-sold concrete block | 6.51 | 6.78 | 6.67 | 6.56 | 6.71 | 6.65 |
| | Prototype brick blocks | 7.67 | 7.54 | 7.26 | 7.41 | 7.95 | 7.57 |
| | containing 200 g native straw | 7.54 | 7.21 | 7.49 | 7.48 | 7.68 | 7.48 |
| 7 days | containing 300 g native straw | 7.65 | 7.46 | 7.12 | 7.61 | 7.43 | 7.45 |
| | containing 400 g native straw | 7.29 | 7.67 | 7.43 | 7.36 | 7.51 | 7.45 |
| | containing 500 g native straw | 7.31 | 7.01 | 7.47 | 7.25 | 7.03 | 7.21 |
| | Prototype brick blocks | 7.56 | 7.49 | 7.67 | 7.51 | 7.51 | 7.55 |
| | containing 200 g native straw | 7.36 | 7.35 | 7.22 | 7.45 | 7.61 | 7.40 |
| 14 days | containing 300 g native straw | 7.25 | 7.36 | 7.23 | 7.27 | 7.54 | 7.33 |
| | containing 400 g native straw | 7.32 | 7.36 | 7.39 | 7.26 | 7.30 | 7.33 |
| | containing 500 g native straw | 7.24 | 7.42 | 7.19 | 7.26 | 7.16 | 7.25 |
| | Prototype brick blocks | 6.93 | 7.36 | 7.73 | 7.89 | 7.54 | 7.49 |
| | containing 200 g native straw | 7.44 | 7.57 | 7.66 | 7.65 | 7.61 | 7.59 |
| 28 days | containing 300 g native straw | 7.28 | 7.57 | 7.61 | 7.42 | 7.61 | 7.50 |
| | containing 400 g native straw | 7.51 | 7.58 | 7.38 | 7.45 | 7.35 | 7.45 |
| | containing 500 g native straw | 7.46 | 7.41 | 7.30 | 7.30 | 7.58 | 7.41 |

**Table 5.** The results of the weight of brick blocks containing 200, 300, 400 or 500 g of native rice straw in place of sand compared to the prototype bricks.

| Curing Time | Brick Type | Weight of Brick (kg) | | | | | Average |
|---|---|---|---|---|---|---|---|
| | | 1 | 2 | 3 | 4 | 5 | |
| - | Market-sold concrete block | 6.51 | 6.78 | 6.67 | 6.56 | 6.71 | 6.65 |
| | Prototype brick blocks | 7.67 | 7.54 | 7.26 | 7.41 | 7.95 | 7.57 |
| | containing 200 g native straw | 7.51 | 7.68 | 7.70 | 7.62 | 7.69 | 7.64 |
| 7 days | containing 300 g native straw | 7.58 | 7.55 | 7.57 | 7.49 | 7.70 | 7.58 |
| | containing 400 g native straw | 7.61 | 7.80 | 7.81 | 7.63 | 7.02 | 7.57 |
| | containing 500 g native straw | 7.43 | 7.12 | 7.17 | 7.01 | 7.22 | 7.19 |
| | Prototype brick blocks | 7.56 | 7.49 | 7.67 | 7.51 | 7.51 | 7.55 |
| | containing 200 g native straw | 6.91 | 7.69 | 8.05 | 7.41 | 7.50 | 7.51 |
| 14 days | containing 300 g native straw | 6.48 | 7.89 | 7.77 | 7.73 | 7.53 | 7.48 |
| | containing 400 g native straw | 7.31 | 7.57 | 7.69 | 7.73 | 7.07 | 7.47 |
| | containing 500 g native straw | 6.88 | 6.92 | 6.76 | 7.05 | 7.08 | 6.94 |
| | Prototype brick blocks | 6.93 | 7.36 | 7.73 | 7.89 | 7.54 | 7.49 |
| | containing 200 g native straw | 7.51 | 7.48 | 7.50 | 7.64 | 7.53 | 7.53 |
| 28 days | containing 300 g native straw | 7.61 | 7.43 | 7.63 | 7.42 | 7.37 | 7.49 |
| | containing 400 g native straw | 7.07 | 7.48 | 7.57 | 7.42 | 7.63 | 7.43 |
| | containing 500 g native straw | 7.30 | 7.27 | 6.64 | 6.61 | 6.60 | 6.88 |

**Table 6.** The results of the weight of brick blocks containing 200, 300, 400 or 500 g of native rice straw in place of stone compared to the prototype bricks.

| Curing Time | Block Type | Weight of Brick (kg) | | | | | Average |
|---|---|---|---|---|---|---|---|
| | | 1 | 2 | 3 | 4 | 5 | |
| - | Market-sold concrete block | 6.51 | 6.78 | 6.67 | 6.56 | 6.71 | 6.65 |
| | Prototype brick blocks | 7.67 | 7.54 | 7.26 | 7.41 | 7.95 | 7.57 |
| | containing 200 g native straw | 7.03 | 6.99 | 7.22 | 7.24 | 6.81 | 7.06 |
| 7 days | containing 300 g native straw | 7.29 | 6.93 | 6.83 | 7.12 | 6.89 | 7.01 |
| | containing 400 g native straw | 6.00 | 6.97 | 7.57 | 6.99 | 7.17 | 6.94 |
| | containing 500 g native straw | 7.15 | 6.56 | 6.75 | 6.86 | 6.43 | 6.75 |

**Table 6.** *Cont.*

| Curing Time | Block Type | Weight of Brick (kg) | | | | | Average |
| --- | --- | --- | --- | --- | --- | --- | --- |
| | | 1 | 2 | 3 | 4 | 5 | |
| 14 days | Prototype brick blocks | 7.56 | 7.49 | 7.67 | 7.51 | 7.51 | 7.55 |
| | containing 200 g native straw | 7.32 | 8.05 | 8.03 | 6.73 | 6.67 | 7.36 |
| | containing 300 g native straw | 7.37 | 7.01 | 6.98 | 7.08 | 7.03 | 7.09 |
| | containing 400 g native straw | 6.83 | 7.05 | 6.84 | 7.32 | 6.98 | 7.00 |
| | containing 500 g native straw | 6.26 | 7.27 | 6.74 | 6.66 | 6.85 | 6.76 |
| 28 days | Prototype brick blocks | 6.93 | 7.36 | 7.73 | 7.89 | 7.54 | 7.49 |
| | containing 200 g native straw | 7.51 | 7.34 | 7.56 | 7.22 | 7.12 | 7.35 |
| | containing 300 g native straw | 7.09 | 7.32 | 7.27 | 7.00 | 6.98 | 7.13 |
| | containing 400 g native straw | 6.40 | 6.45 | 6.47 | 6.60 | 6.97 | 6.58 |
| | containing 500 g native straw | 6.40 | 6.45 | 6.47 | 6.60 | 6.97 | 6.58 |

The results of the weight of bricks made with 200, 300, 400 or 500 g of native rice straw in place of cement in the 5 samples per mixture show that the weight tends to decrease. In the mixture with less native rice straw content, the weight was higher than in the mixture with higher native rice straw content. The weight value compared to the original brick block is not significantly different. However, it can be seen that the bricks that are sold in the market have a lighter weight.

From the results of the weight of bricks mixed with native rice straw in the amount of 200, 300, 400 or 500 g to replace the sand in the amount of 5 samples per mixture, it can be seen that the weight tends to decrease with an increase in straw. In the mixture with less native rice straw content, the brick block's weight was higher than in the mixture with a higher native rice straw content. However, brick blocks that are sold in the market are still lighter. The decrease in weight is because in the raw material mixing, the ratio of 1:3:5 (cement:sand:stone) was used. As the amount of straw in the straw/cement mixture increased, the weight of the block decreased, as straw weighs less than cement.

The results of the weight of 5 samples of bricks mixed with native rice straw in the amount of 200, 300, 400 or 500 g to replace stone show that the weight tends to decrease with more straw. In the mixture with less native rice straw content, the weight was higher than in the mixture with higher native rice straw content and the compression strength compared to the prototype brick block was different because, in the raw material mixing, the ratio of 1:3:5 (cement:sand:stone) was used. There were 5 parts of stone compared to cement, in the stone mixture ratio. The bricks that are sold in the market are lighter.

*4.3. Results of Heat Absorption*

Prototype brick blocks and brick blocks containing 200, 300, 400 or 500 g of native rice straw in ratio to cement, stone and sand were incubated for a period of 7, 14 or 28 days. A mercury thermometer was used to test the heat absorption. There are two types of temperature measurements. The first involves measuring inside the individual brick blocks by bringing the bricks blocks of each ingredient together to form a rectangle and covering the top with bricks. Three thermometers are placed in the box, and the temperature is calculated by finding the mean temperature at each interval. The second method is to measure the outside temperature by placing three thermometers on the exterior surface of the box and calculating the average temperature at each interval, as shown in Figure 5. The experiments were conducted from 09:30 to 19:00, and the values were recorded every 30 min. The internal and external temperatures of commercially available bricks were also measured in the same way. The obtained values were used to find the mean every two hours, as shown in Tables 7–9.

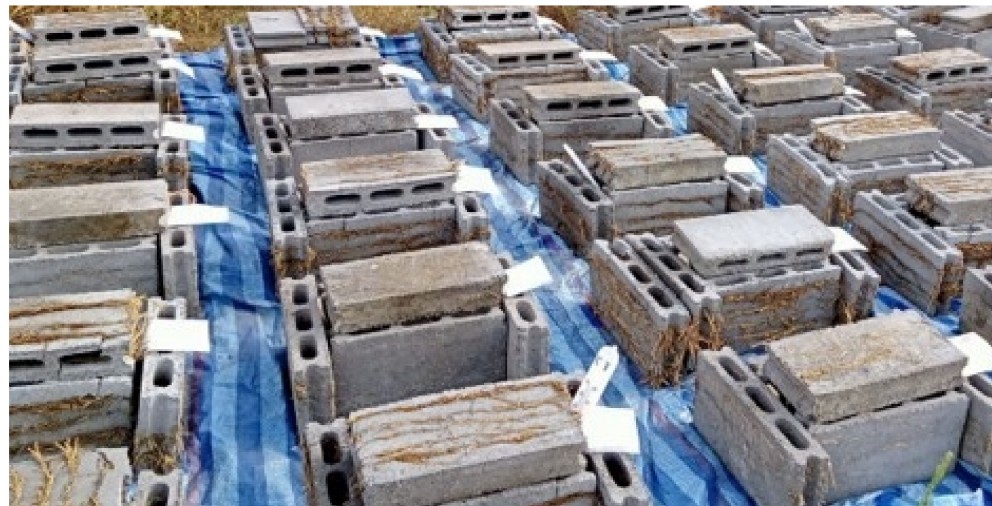

**Figure 5.** Heat absorption experiment.

**Table 7.** The results of heat absorption of brick blocks containing 200, 300, 400 or 500 g of native rice straw in place of cement compared to the prototype bricks.

| Curing Time | Brick Type | Temperature (°C) | | | | | | | |
|---|---|---|---|---|---|---|---|---|---|
| | | 09:30–11:30 | | 12:00–14:00 | | 14:30–16:30 | | 17:00–19:00 | |
| | | External | Internal | External | Internal | External | Internal | External | Internal |
| - | Market-sold concrete block | 33 | 25 | 40 | 30 | 39 | 32 | 25 | 28 |
| 7 days | Prototype brick blocks | 33 | 24 | 42 | 31 | 41 | 34 | 27 | 28 |
| | containing 200 g native straw | 33 | 22 | 42 | 32 | 41 | 36 | 27 | 28 |
| | containing 300 g native straw | 33 | 24 | 42 | 32 | 41 | 35 | 27 | 28 |
| | containing 400 g native straw | 33 | 23 | 42 | 31 | 41 | 35 | 27 | 27 |
| | containing 500 g native straw | 33 | 26 | 42 | 34 | 41 | 38 | 27 | 32 |
| 14 days | Prototype brick blocks | 33 | 24 | 42 | 31 | 41 | 33 | 27 | 28 |
| | containing 200 g native straw | 33 | 22 | 42 | 30 | 41 | 33 | 27 | 28 |
| | containing 300 g native straw | 33 | 24 | 42 | 31 | 41 | 34 | 27 | 28 |
| | containing 400 g native straw | 33 | 22 | 42 | 31 | 41 | 35 | 27 | 30 |
| | containing 500 g native straw | 33 | 22 | 42 | 32 | 41 | 37 | 27 | 27 |
| 28 days | Prototype brick blocks | 33 | 22 | 42 | 31 | 41 | 33 | 27 | 27 |
| | containing 200 g native straw | 33 | 23 | 42 | 32 | 41 | 35 | 27 | 28 |
| | containing 300 g native straw | 33 | 25 | 42 | 34 | 41 | 37 | 27 | 28 |
| | containing 400 g native straw | 33 | 25 | 42 | 32 | 41 | 36 | 27 | 30 |
| | containing 500 g native straw | 33 | 25 | 42 | 32 | 41 | 34 | 27 | 28 |

**Table 8.** The results of heat absorption of brick blocks containing 200, 300, 400 or 500 g of native rice straw in place of sand compared to the prototype bricks.

| Curing Time | Brick Type | Temperature (°C) | | | | | | | |
|---|---|---|---|---|---|---|---|---|---|
| | | 09:30–11:30 | | 12:00–14:00 | | 14:30–16:30 | | 17:00–19:00 | |
| | | External | Internal | External | Internal | External | Internal | External | Internal |
| - | Market-sold concrete block | 33 | 25 | 40 | 30 | 39 | 32 | 25 | 28 |
| 7 days | Prototype brick blocks | 33 | 24 | 42 | 31 | 41 | 34 | 27 | 28 |
| | containing 200 g native straw | 33 | 22 | 42 | 31 | 41 | 32 | 27 | 26 |
| | containing 300 g native straw | 33 | 23 | 42 | 32 | 41 | 35 | 27 | 28 |
| | containing 400 g native straw | 33 | 24 | 42 | 31 | 41 | 33 | 27 | 27 |
| | containing 500 g native straw | 33 | 23 | 42 | 29 | 41 | 31 | 27 | 28 |
| 14 days | Prototype brick blocks | 33 | 24 | 42 | 31 | 41 | 33 | 27 | 28 |
| | containing 200 g native straw | 33 | 23 | 42 | 30 | 41 | 32 | 27 | 28 |
| | containing 300 g native straw | 33 | 23 | 42 | 31 | 41 | 35 | 27 | 28 |
| | containing 400 g native straw | 33 | 23 | 42 | 31 | 41 | 33 | 27 | 28 |
| | containing 500 g native straw | 33 | 24 | 42 | 32 | 41 | 33 | 27 | 27 |
| 28 days | Prototype brick blocks | 33 | 22 | 42 | 31 | 41 | 33 | 27 | 27 |
| | containing 200 g native straw | 33 | 23 | 42 | 31 | 41 | 33 | 27 | 25 |
| | containing 300 g native straw | 33 | 23 | 42 | 31 | 41 | 34 | 27 | 28 |
| | containing 400 g native straw | 33 | 24 | 42 | 33 | 41 | 35 | 27 | 29 |
| | containing 500 g native straw | 33 | 24 | 42 | 31 | 41 | 33 | 27 | 26 |

**Table 9.** The results of heat absorption of brick blocks containing 200, 300, 400 or 500 g of native rice straw in place of stone compared to the prototype bricks.

| Curing Time | Brick Type | Temperature (°C) | | | | | | | |
|---|---|---|---|---|---|---|---|---|---|
| | | 09:30–11:30 | | 12:00–14:00 | | 14:30–16:30 | | 17:00–19:00 | |
| | | External | Internal | External | Internal | External | Internal | External | Internal |
| - | Market-sold concrete block | 33 | 25 | 40 | 30 | 39 | 32 | 25 | 28 |
| 7 days | Prototype brick blocks | 33 | 24 | 42 | 31 | 41 | 34 | 27 | 28 |
| | containing 200 g native straw | 33 | 22 | 42 | 31 | 41 | 34 | 27 | 27 |
| | containing 300 g native straw | 33 | 24 | 42 | 31 | 41 | 31 | 27 | 27 |
| | containing 400 g native straw | 33 | 23 | 42 | 32 | 41 | 35 | 27 | 28 |
| | containing 500 g native straw | 33 | 24 | 42 | 31 | 41 | 33 | 27 | 29 |

**Table 9.** *Cont.*

| Curing Time | Brick Type | Temperature (°C) | | | | | | | |
|---|---|---|---|---|---|---|---|---|---|
| | | 09:30–11:30 | | 12:00–14:00 | | 14:30–16:30 | | 17:00–19:00 | |
| | | External | Internal | External | Internal | External | Internal | External | Internal |
| 14 days | Prototype brick blocks | 33 | 24 | 42 | 31 | 41 | 33 | 27 | 28 |
| | containing 200 g native straw | 33 | 23 | 42 | 31 | 41 | 34 | 27 | 26 |
| | containing 300 g native straw | 33 | 25 | 42 | 33 | 41 | 34 | 27 | 28 |
| | containing 400 g native straw | 33 | 24 | 42 | 32 | 41 | 33 | 27 | 26 |
| | containing 500 g native straw | 33 | 25 | 42 | 34 | 41 | 33 | 27 | 27 |
| 28 days | Prototype brick blocks | 33 | 22 | 42 | 31 | 41 | 33 | 27 | 27 |
| | containing 200 g native straw | 33 | 24 | 42 | 30 | 41 | 32 | 27 | 27 |
| | containing 300 g native straw | 33 | 24 | 42 | 30 | 41 | 32 | 27 | 27 |
| | containing 400 g native straw | 33 | 24 | 42 | 32 | 41 | 34 | 27 | 27 |
| | containing 500 g native straw | 33 | 25 | 42 | 32 | 41 | 33 | 27 | 27 |

The results of the heat transfer coefficient of brick blocks containing 200, 300, 400 or 500 g of native rice straw in place of cement in 5 samples show that the heat absorption of brick blocks varies according to the external temperature but generally produces a temperature that is 5–10 degrees Celsius cooler than the market brick. However, between 17:00 and 19:00, the temperature inside the brick blocks was higher than the temperature outside. This is because the exothermic effect of the brick blocks is slower than the outside temperature.

The results of the heat transfer coefficient of brick blocks containing 200, 300, 400 or 500 g of native rice straw in place of sand in 5 samples show that the heat absorption of brick blocks varies according to the external temperature but generally produces a temperature that is 5–10 degrees Celsius cooler than the market brick. However, between 17:00 and 19:00, the temperature inside the brick blocks was higher than the temperature outside. This is because the heat dissipation of brick blocks is slower than the outside temperature.

The results of the heat absorption of brick blocks containing 200, 300, 400 or 500 g of native rice straw in place of stone in 5 samples show that the heat absorption of brick blocks varies according to the external temperature but generally produces a temperature that is 5–10 degrees Celsius cooler than the market brick. However, between 17:00 and 19:00, the temperature inside the brick blocks was higher than the temperature outside. This is because the exothermic effect of brick blocks is slower than the outside temperature.

*4.4. Price Test Results*

The price of bricks and blocks available in the market in December 2021 was 7.00 baht ($0.20 USD) per block, after deducting labor costs (1 USD = 35.00 baht). Based on the Thai minimum wage in 2022 of 300 baht per day, the research team was able to produce 1000 cubes per day; with a wage per bale of 0.30 baht, operating costs of 10 percent and a profit of 15 percent, the cost of producing bricks and blocks that are sold in the market is 5.30 baht per block.

The price test of the bricks containing the mixture of native rice straw and the prototype brick, which measures 7 × 19 × 39 cm, were priced at 3.17 baht ($0.09 USD) per brick (Table 10).

**Table 10.** Price test results of block bricks mixed with native rice straw compared with prototype bricks.

| Type of Brick Block | | Price (Baht/kg) | | | | |
|---|---|---|---|---|---|---|
| | | Cement | Sand | Stone | Native Rice Straw | Price/Brick |
| - | Market-sold concrete block | - | - | - | - | 5.30 |
| - | Prototype brick blocks | 36.40 | 3.90 | 6.75 | 0.00 | 3.14 |
| Replace cement | containing 200 g native straw | 35.67 | 3.90 | 6.75 | 1.30 | 3.17 |
| | containing 300 g native straw | 35.31 | 3.90 | 6.75 | 1.95 | 3.19 |
| | containing 400 g native straw | 34.94 | 3.90 | 6.75 | 2.60 | 3.21 |
| | containing 500 g native straw | 34.58 | 3.90 | 6.75 | 3.25 | 3.23 |
| Replace sand | containing 200 g native straw | 36.40 | 3.82 | 6.75 | 3.00 | 3.33 |
| | containing 300 g native straw | 36.40 | 3.78 | 6.75 | 4.50 | 3.43 |
| | containing 400 g native straw | 36.40 | 3.74 | 6.75 | 6.00 | 3.53 |
| | containing 500 g native straw | 36.40 | 3.71 | 6.75 | 7.50 | 3.62 |
| Replace stone | containing 200 g native straw | 36.40 | 3.90 | 6.62 | 4.50 | 3.43 |
| | containing 300 g native straw | 36.40 | 3.90 | 6.55 | 6.75 | 3.57 |
| | containing 400 g native straw | 36.40 | 3.90 | 6.48 | 9.00 | 3.72 |
| | containing 500 g native straw | 36.40 | 3.90 | 6.41 | 11.25 | 3.86 |

From the price test results, it was found that bricks mixed with a higher quantity of native rice straw saw an increase in the price due to the high price of native rice straw. However, this price is lower than bricks that are sold in the market.

## 5. Discussion and Conclusions

### 5.1. Summary of Compression Strength Results

From the results of the compression strength test, it can be seen that the bricks mixed with 200 g of native rice straw instead of cement could withstand the maximum weight of 1.26 kg/cm$^2$. Adding more native rice straw will reduce the load bearing of bricks containing the native rice straw mixture accordingly. This is consistent with previous research [3] on the development and study of compression strength of rice straw concrete blocks; when the amount of rice straw increases, the compressive strength is reduced [6,7,14].

### 5.2. Summary of Weight of Bricks

From the results of the weight of the bricks, it can be seen that the bricks mixed with 500 g of native rice straw instead of stone will weigh the least at 6.58 kg, but they are unable to bear weight due to the large amount of native rice straw. Therefore, bricks mixed with 200 g of native rice straw instead of cement will reduce the weight of the brick. This is consistent with previous findings, which indicated that an increased amount of straw reduces the weight of the brick blocks [15,16].

### 5.3. Results of Heat Absorption

This study compared the heat transfer coefficient by comparing the incubation period of 28 days from 12:00–14:00, which is the time when the outdoor temperature is highest. Bricks mixed with 200 g of native rice straw instead of stone can reduce the temperature by 12 degrees Celsius, more than any other combination. However, these bricks are unable to bear weight due to the large amount of native rice straw. Therefore, a better choice for production would be bricks mixed with 200 g of native rice straw in place of cement, which can reduce the temperature by 10 degrees Celsius but have a better weight compression strength. The best heat transfer properties were the block bricks containing 200 g of native rice straw instead of cement, which was consistent with the findings of previous studies. This indicated that the thermal conductivity of the rice straw cement block will decrease as the percentage of rice straw fiber increases [13,17,18].

*5.4. Summary of the Price Test Results*

From the results, comparing the mixture of native rice straw to replace cement, sand and stone for the objectives, it can be concluded that the block with the best properties in all four aspects is the block containing a 200 g mixture of native rice straw replacing cement. These bricks had a weight of 1.26 kg/cm$^2$. The weight per bale is 7.59 kg per bale. These bricks can reduce the heat by 10 degrees Celsius and have a price per piece of 3.17 baht, which is a lower production cost than bricks in the market. As such, these bricks are a possibility as an alternative to use or to produce for further sale [19,20].

In conclusion, bricks mixed with native rice straw have good compression strength, good heat transfer coefficients and a low price. Native rice straw mixed bricks are attractive for energy saving when used as wall insulation and appropriate for environmentally friendly construction [12,19].

**Author Contributions:** Conceptualization, S.T.; writing—original draft preparation, S.T. and S.H.; methodology, S.T.; formal analysis, S.T.; investigation, S.T.; data curation, S.T. and S.H.; visualization, S.T.; writing—review and editing, S.T.; supervision, S.T. All authors have read and agreed to the published version of the manuscript.

**Funding:** This research received no external funding.

**Acknowledgments:** The authors gratefully acknowledge Chiang Mai Rajabhat University, Thailand.

**Conflicts of Interest:** The authors declare no conflict of interest.

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
