# Peer review of "Investigation of Compression Strength and Heat Absorption of Native Rice Straw Bricks for Environmentally Friendly Construction"

_sustainability, doi:10.3390/su141912229_

Round 1

Reviewer 1 Report

Review

Manuscript Number: SUSTAINABILITY-1842705

Title:Investigation of Compression Strength and Heat Transfer Coefficient of Native Rice Straw Brick for Environmentally Friendly Construction

Article Type: Research Paper

The submitted work presents results of an experimental study regarding the efficiency of native rice straw mixed bricks. The experimental study includes results considering the compression strength the heat tranfer coefficient and the weight of the examined bricks. A comparative analysis between the examined bricks and the market-sold concrete blocks has been also carried out.

In Reviewers' opinion the level of the used English is considered below average. Several syntax, grammar and spelling mistakes were detected throughout the text that create several misunderstandings. Due to the low level of the used English the results and the contribution of this research in the literature are not understandable by the readers. For the aforementioned reasons, the submitted work can not be accepted in the present form.

The main purpose of the submitted work with code name SUSTAINABILITY-1842705 is to investigate the efficiency of native rice straw as an additive in brick walls and to produce a more environmental-friendly constructive material. Especially, native rice straw species from three different Origins were used to replace cement, sand and stone in the amount of 20, 30, 40, 50 g. The experimental study includes results considering the compression strength, the heat transfer coefficient, and the weight of the purposed bricks. A comparative analysis between the examined bricks and market-sold concrete blocks has been also carried out.
Even if the topic of this paper is quite interesting, the way the text is written creates several misunderstandings regarding the scope and the originality of this research. As I had been already mentioned the submitted paper is not well written. The level of the used English is low as several syntax, grammar and spelling mistakes are detected throughout the text.
According to the experimental results (see table 4.1.1) the compression strength of brick blocks containing native rice straw in a ratio of 20g instead of 20g concrete is almost the same with the prototype brick clocks but quite lower compared with the compression strength of the market sold concrete blocks. According to tables 4.1.2 and 4.1.3 the examined bricks were not in a condition that can be tested, so no clear conclusions can be drawn regarding the compression strength of the proposed cases.
Considering the weight and the heat transfer coefficient of the proposed brick blocks, most of the examined cases presented higher weight and similar behavior compared to the market sold concrete blocks. Finally, concerning the competitive pricing analysis, it would be helpful and useful if the Authors also follow the international variation standards and not only local values.
For all the above reasons the submitted work can not be accepted in the present form. However, the Reviewer would like to encourage the authors to continue their experimental research as their approach is very interesting and useful to research.

Author Response

Dear Reviewer 2, 

Thank you very much for your valuable suggestions. I revised my manuscript according to your suggestions. Please see attached file. 

Best regards, 

Suchat Tachaudomdach 

Reviewer 2 Report

The article deals with the interesting topic of biobased construction using agricultural waste. However, it needs some improvements:

1- Add at least one reference to justify the first two sentences of the introduction “Nowadays, construction often uses recycled materials from agriculture as they are 27 the most cost-effective and efficient. This leads to a reduction in waste and a reduction in 28 environmental impact.”

2- Most references to related research (section 2.6) are not very recent. Some more recent references to similar studies should be added. This research's novelty compared to the others should also be stated.

3- Section 4 is misnamed as discussion and should be called “Results”

4- Add images of the resulting straw blocks. The study only shows pictures of the conventional concrete blocks

5- The results of section 4.3 do not show heat transfer coefficients but merely temperature differences. The term temperature coefficient should be removed if the actual heat transfer coefficients are not added. Adding the coefficients would be the optimal solution.

Author Response

Dear Reviewer 2:

Thank you so much for valuable suggestions. I revised my manuscript according to your suggestions.  I highlighted with blue color. Please see attached file.

Thank you very much again for valuable suggestions.

Best regards,

Round 2

Reviewer 1 Report

I would like to congratulate the Authors for the excellent work that they have done to improve the quality of their submitted work and to thank them for taking account all the recommendations that this Reviewer has done. The quality of this work has been improved. However, this Reviewer suggests some extra trivial recommendations to improve the overall organization of this paper. The Authors can find the recommendations in the attached file highlighted with yellow color. So, this Reviewer believes that the submitted paper should be accepted after minor revision.

Author Response

Dear Reviewer 1:

Thank you so much for valuable suggestions. I revised my manuscript according to your suggestions.  I highlighted with green color. Please see attached file.

Thank you very much again for valuable suggestions.

Best regards,

Reviewer 2 Report

You made enough changes for me to accept the manuscript. I would recommend doing a more thorough literature review in the future.

Author Response

Dear Reviewer 2:

Thank you so much for valuable suggestions. I revised my manuscript according to your suggestions.  I highlighted with blue color. Please see detail below.

Thank you very much again for valuable suggestions.

Best regards,
